# Global change factors differ in effect when acting alone and in a multi-factor background

Rebecca Rongstock[1,2,3], Huiying Li[1,2,3], Anika Lehmann [1,2], Anja Wulf[1] &
Matthias C. Rillig [1,2] ✉

The presence of multiple global change factors affects most ecosystems. Urban soils face stressors like heat, drought, road salt, nitrogen deposition, surfactants, and microplastics. Given that combined factors of global change have shown unpredictable effects, we here ask which individual factors have particularly negative effects in multifactorial contexts. We explore this through a subtractive design, comparing single-factor treatments (addition) to treatments where a specific factor is removed (subtraction). The results vary from predominantly negative, positive, to mixed effects. However, removing these factors from a multi-factor context generally improves soil properties and biological processes. Resource related factors enhance microbial activity individually but show no such benefit in multi-factor scenarios. Our findings highlight that the combined effects of factors often differ from their individual impacts. In restoration, priority should be given to mitigating factors with the strongest negative influence in multi-stressor contexts, rather than targeting those with significant isolated effects.

Human influence on ecosystems is omnipresent and multi-faceted. Among the general anthropogenic influences relevant to the soil ecosystem are urbanization, agriculture, globalization and industrial activities[1,2]. The resulting factors are diverse and can be of biological, chemical and physical nature. They include artificial light at night, warming, soil compaction and sealing, the introduction of plastics, nutrient enrichment, drought, elevated $CO_2$, heavy metals, salinity, biocides and the spread of invasive species[3,4].

Individual anthropogenic factors can reinforce or attenuate each other. Warming without water limitation usually leads to increased plant biomass, while additional drought reverses the effect to the negative, as warming promotes evaporation of water from plants and soil[5]. This example demonstrates, as shown in many other studies, how one global change factor can amplify, reduce or reverse the effect of other factors[6], but it is unclear whether such interactions between two factors are more likely to be antagonistic or synergistic[5,7]. Global change consists of numerous interacting factors. Nevertheless, in

research practice, experiments typically deal with just one or two factors[8]. In an experiment in which global change factors were randomly selected from a pool of 10 to create a gradient with an increasing number of factors, it was demonstrated that an increasing number of anthropogenic factors can cause increasing directional changes in soil properties, function, and soil biodiversity[8]. Other studies using factor gradients have drawn similar conclusions for different plant and soil systems. The main finding of studies applying factor gradients is that an increasing number of global change factors often has stronger negative effects than predicted by the addition of individual factors[9–12]. While factor gradients can be applied to ask whether the effect of multiple interacting factors can be predicted by single factor effects, we here propose a design that provides a step toward a more mechanistic understanding, focusing on the effect of individual factors in isolation and within a multifactorial context with implications for restoration efforts. This enables us to identify factors with the most dramatic impacts when combined with other factors of global

[1]Freie Universität Berlin, Institute of Biology, Berlin, Germany. [2]Berlin-Brandenburg Institute of Advanced Biodiversity Research, Berlin, Germany. [3]These authors contributed equally: Rebecca Rongstock, Huiying Li. ✉e-mail: rillig@zedat.fu-berlin.de

change, and this is an important aspect thus far not covered by any of the studies using factor gradient approaches.

The presence or intensity of global change factors hinder restoration efforts making it necessary to reduce the number of global change factors. This raises the question of which global change factors should be reduced or eliminated first. While restoration approaches have often focussed on vegetation, soil organisms and processes have been less frequently considered[13]. An unconventional but emerging field for conservation efforts is urban ecosystems, with their particular intensity and diversity of global change factors. In urban areas, global change factors occur in specific combinations and with great heterogeneity, creating unique difficulties for urban soil restoration[14]. Prioritizing the elimination of factors of global change with a particularly strong effect in combination with other factors could therefore be a promising approach for restoration efforts.

In our experiment, we use a subtractive design to test effects on soil processes and functions influenced by global change factors that frequently co-occur in urban environments: warming, drought, salinity, nitrogen deposition, car tire abrasion, and surfactant. We compare the difference in effect sizes and directions between control and single factor addition with all factors and all factors minus one single factor. In doing so, we test the following three hypotheses: (1) Due to the nature of the chosen global change factors there will be positive, negative and neutral single factor effects. Some of the chosen global change factors are resources or stressors, depending on the soil function or process in focus. (2) Higher order factor combinations will negatively affect soil functions and processes irrespective of the sign and magnitude of the single global change factor treatments. (3) Effect size magnitudes and directions will differ depending on whether a factor is added to control soil or whether the factor is removed from the background of the multiple global change factors because of synergistic and antagonistic effects of factor combinations. In the context of restoration, it is important to take such differences into account when deciding which anthropogenic factors should be prioritized for reduction or removal efforts.

## Results

In the following, the effects of the single factors compared to the control, the effects of the factor in the background of the multiple factors and the correspondence between these effect sizes and directions will be presented for the response variables related to microbial activity (enzyme activity and decomposition), and soil structure and properties (mean weight diameter, water stable aggregates, and water drop penetration time). In general, salinity caused negative effects on soil functions and processes as a single factor, while in the background of the multiple factors, removed salinity improved functions and processes compared to the combination of all global change factor treatments (p-values shown in Table 1, $R^2$ values for redundancy analysis and variance partitioning in Supplementary table 2). Warming effects were mostly positive when comparing single factor treatments to the control, but also positive when removed from the background of the multiple factors. Drought had negative, positive, and neutral effects in both single factor addition and in the background of the multiple factors. For microplastic and surfactant, only trends could be detected. Warming and drought were similar to each other in effect size and direction and salinity effects showed more similarity to drought effects (Supplementary Fig. 1).

### Single factor effects compared to control

Salinity had negative effects on β-D-glucosidase activity, acid phosphatase activity, and decomposition (Fig. 1a, e, b, f, d, h). The percentage of water stable aggregates (WSA) decreased in the salinity treatments (Fig. 2b, e) and mean weight diameter (MWD) was unaffected (Fig. 2a, d). Warming increased N-acetyl-β-glucosaminidase activity (Fig. 1c, g) and WSA. The MWD was negatively affected by the addition of drought. Additionally, drought increased the time for a droplet of water to infiltrate (Fig. 2c, f). However, drought increased the decomposition rate (Fig. 1d, h). Microplastics and surfactants showed a trend to increase β-D-glucosidase activity (Fig. 1a, e). N deposition had no measurable effects.

### Effects of all global change factors compared to control and single factors

The combination of all global change factors consistently showed strong negative effects compared to the control. The effects of single factors were less intense than the effect of all factors in combination. There was one exception regarding the response variable water drop penetration time. Here, the effect of the single factor drought was not significantly different from the combination of all factors.

### Effects of the removal of single factors from the background of all global change factors

When factors are subtracted from the background of all other global change factors, the effect can be influenced by the factor itself, its interaction with other factors or unexplained variability (biological noise or measurement errors). In order to differentiate between these causes, and to pinpoint what part of the variability is not due to noise or measurement error, we expressed these sources of variance as shared variance and residual variance, the latter capturing

**Table 1 | P-values and significance test from two-sided one-way ANOVA followed by Tukey's post-hoc test**

| | Factors | N-acetyl-glucosaminidase activity | β-glucosidase activity | Phosphatase activity | Decomposition rate | WSA | MWD | Water drop penetration time |
|---|---|---|---|---|---|---|---|---|
| **Control +** | Salinity | 0.3445 | < 0.001 *** | < 0.001 *** | < 0.001 *** | < 0.001 *** | 0.689 | 1.000 |
| | N deposition | 0.9711 | 0.5051 | 0.996 | 1.000 | 0.6770 | 0.482 | 1.000 |
| | Drought | 0.8570 | 0.6269 | 0.739 | < 0.001 *** | 0.4707 | < 0.001 *** | < 0.001 *** |
| | Warming | 0.0245 * | 0.3106 | 0.976 | < 0.001 *** | 0.0205 * | 0.886 | 1.000 |
| | Microplastic | 0.8383 | 0.0670 | 0.995 | 0.563 | 0.9996 | 0.520 | 1.000 |
| | Surfactant | 0.9418 | 0.0876 | 0.779 | 0.981 | 0.9995 | 0.968 | 1.000 |
| **All -** | Salinity | < 0.001 *** | < 0.001 *** | < 0.001 *** | < 0.001 *** | < 0.001 *** | 1.000 | 0.050 * |
| | N deposition | 0.932 | 1.0000 | 0.988 | 0.607 | 0.404 | 1.000 | 1.000 |
| | Drought | <0.001 *** | 0.00671 ** | < 0.001 *** | < 0.001 *** | 0.110 | < 0.001 *** | < 0.001 *** |
| | Warming | <0.001 *** | <0.001 *** | <0.001 *** | 1.000 | 0.502 | 0.823 | 0.548 |
| | Microplastic | 0.904 | 0.99895 | 0.998 | 0.998 | 0.965 | 0.973 | 1.000 |
| | Surfactant | 0.921 | 1.0000 | 0.998 | 0.733 | 0.442 | 1.000 | 1.000 |

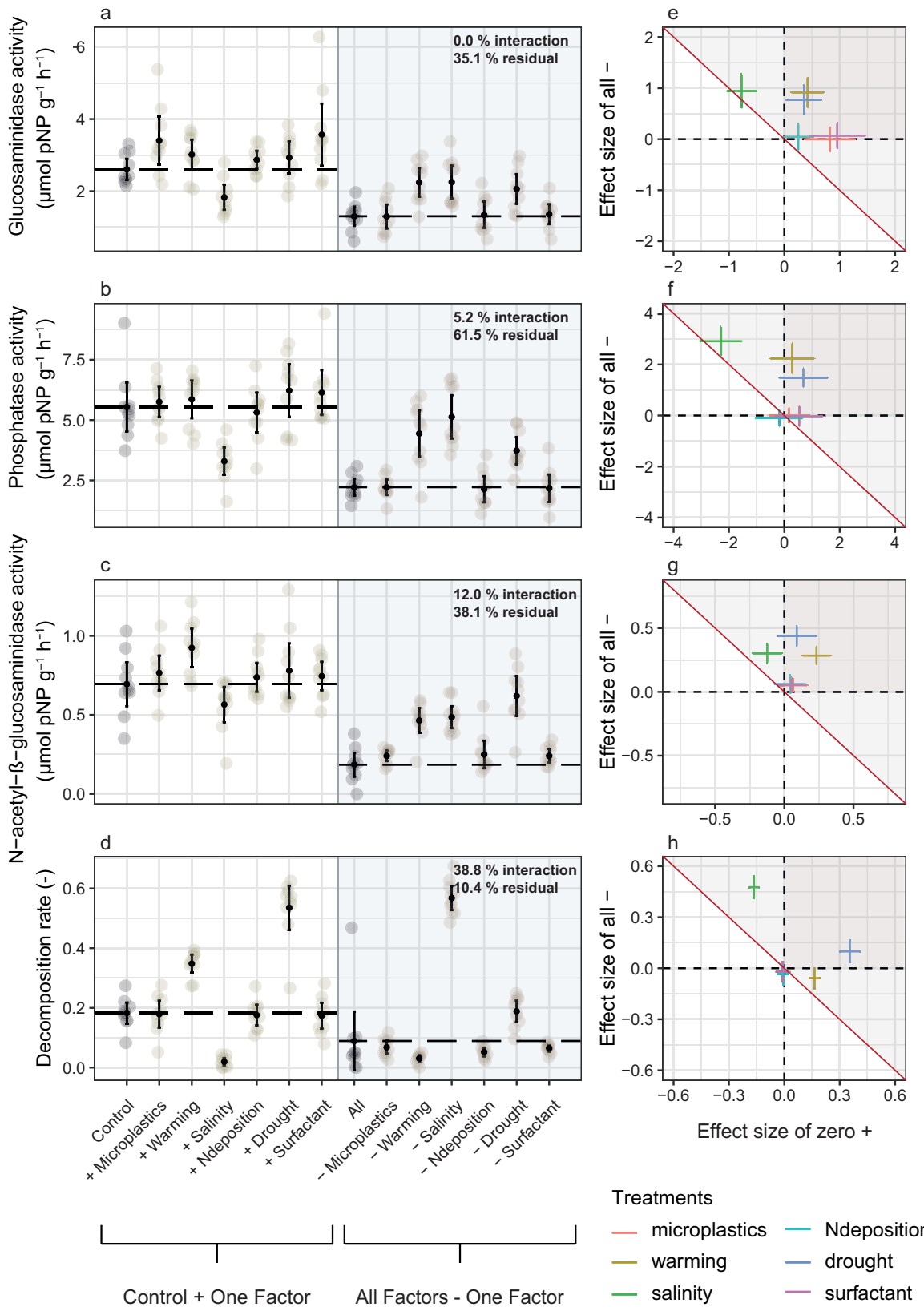

measurement error or biological noise. For enzyme activity, there was generally lower shared interaction among factors of global change (0–12) and a higher proportion of unexplained variance (35.1–61.5%) than in other response variables, indicating a rather strong influence from biological noise on this pattern. In decomposition, MWD, and water drop penetration time, we found shared variance among factors

meaning that the contribution of factors is influenced by interactions with other factors of global change (26.1–39.8%), and less due to biological noise or measurement error for decomposition and MWD (10.4%, 27.0%, respectively). However, the contribution of biological noise or measurement error contributed more to water drop penetration time (46.7%).

**Fig. 1 | Enzyme activities (β-D-glucosidase, N-acetyl-β-glucosaminidase, acid phosphatase) and decomposition affected by multiple factors of global change. a–d** The left part of the graph (white background) shows the control and the single factor treatments. The dashed line shows the mean value of the control plots. Dots reflect the measured values (10 replicates each), marked with a black filled dot is the mean, the 95 % confidence interval is shown by the line passing through the dot. On the right side of the graph, the combination of all global change factors (dashed line as mean), is compared to the treatment combinations, from which each of the single factors is subtracted. Redundancy analysis (RDA) and variance partitioning were performed to quantify the relative contribution of each factor, as well as their interactions, and residual (biological noise or measurement errors), presented in the upper right corner of the panel. **e–h** To visualize the effect

directions of factor addition and subtraction treatments, bootstrapped ($n = 100$) effect size of factor addition were plotted on the $x$ axis and the effect of factor removal on the y-axis. Different factors of global change are represented by different colors. Factor effects on the diagonal red line indicate that the effect of the factor in the multi-factor background has the direction and intensity expected from the single-factor effect. Factor effects in the gray shaded area indicate that a factor in combination with other factors has a more negative effect than expected from the single factor effect. Dark gray areas (upper right) represent positive single factor effects that have a negative effect when combined with other factors (effect size of single factor positive, effect size of factor removal positive as well). Source data are provided as a Source Data file.

Salinity removal from the background of all global change factors significantly improved all response variables except for MWD. In the enzyme activity it independently explained 18.4% of the total variance in N-acetyl-β-glucosaminidase, 13% in β-D-glucosidase activity, and 27.7% in acid phosphatase indicating relatively robust effects. For decomposition, the independently explained variance of salinity was even higher (47.0%). For WSA, 31.4% of the total variance could be explained by salinity alone and 6.7% in water drop penetration time. Multiple global change factor treatments with removed warming showed more enzyme activity compared to treatments with all factors. The factor warming independently explained 12.7–15.9% of the total variance for the activity of the soil enzymes. However, in all other response variables warming explained less than 1% of the total variance. Drought effects in the background of the multiple factors were negative - removing drought improved the performance of all response variables except for WSA. Drought was not the main factor explaining decomposition (1.5%); instead, salinity explained most of the variance. However, it accounted for 44.3% of the variance in MWD, indicating a strong and robust effect on this response variable. With 14.2% of the variance explained, drought had the greatest influence on water drop penetration time. N deposition, surfactant, and microplastic removal did not affect the response variables.

## Correspondence between effects of factors applied singly and in the background of other factors

Comparing effects of factor additions and effects of factors in the background of the multiple factors, some factors of global change showed different effect sizes and directions in single factor addition versus removal from the background of multiple factors.

Salinity as a single factor had mostly negative effects and correspondingly positive effects when it was removed from the background of the multiple factors. On N-acetyl-β-glucosaminidase activity, salinity had no significant effect, but removal from the multiple global change factor background caused an increase in N-acetyl-β-glucosaminidase activity (effect size = 0.30, 95% Confidence Intervals = 0.21, 0.39, $p = < 0.001$) (Table 1, Supplementary information. Salinity treatment decreased acid phosphatase activity compared to the control was (effect size = 2.24, 95% Confidence Intervals = 3.45, −1.47, $p = < 0.001$). An increase of acid phosphatase activity could be detected when comparing all global change factor treatments with the multiple global change factor treatment with removed salinity (effect size = 2.83, 95% Confidence Intervals = 2.07, 3.64, $p = < 0.001$). Salinity also decreased β-D-glucosidase activity with a similar effect size (−0.76, 95% Confidence Intervals = −1.12, −0.38, $p = < 0.001$), as removed salinity from the multiple factor background increased β-D-glucosidase activity (effect size = 0.87, 95% Confidence Intervals = 0.48, 1.31, $p = < 0.001$). Decomposition of wood was negatively affected by salinity (effect size = −1.16, 95% Confidence Intervals = −0.19, −0.13, $p = < 0.001$), whereas removal of salinity from the background of the multiple factors showed a positive, but less pronounced, effect (effect size 0.47, 95% Confidence Intervals = 0.32, 0.53, $p = < 0.001$).

Salinity had neutral or negative effects on soil structure and properties. The percentage of WSA decreased in the salinity treatments (effect size = −11.91, 95% Confidence Intervals = −16.71, −6.08, $p = < 0.001$). Multiple global change factor treatments without salinity had an equally high amount of WSA as the control. No single factor effects of salt on water drop penetration time could be observed because of low values in the control, but in the multiple global change factor background with removed salt, the time required for a drop of water to be absorbed into the soil is extended further (effect size = 17.56, 95% Confidence Intervals = 5.26, 29.63, $p = < 0.050$).

Warming as a single factor had positive or neutral effects, but removing warming from the multiple factor background always increased response variables related to microbial activity. Warming did not affect β-D-glucosidase activity, but in the background of the multiple factors removal of warming increased enzyme activity (effect size = 0.87, 95% Confidence Intervals = 0.46, 1.22, $p = < 0.001$). Similarly, acid phosphatase activity was not affected by addition of warming, but removing it from the multiple global change factor background caused an increase in activity (effect size = 2.14, 95% Confidence Intervals = 1.18, 2.88, $p = < 0.001$). Addition of warming positively affected N-acetyl-β-glucosaminidase activity (effect size = 0.23, 95% Confidence Intervals = 0.08, 0.38, $p = < 0.0245$). Removal of warming from the multiple factor background also resulted in an increase of N-acetyl-β-glucosaminidase activity (effect size = 0.28, 95% Confidence Intervals = 0.17, 0.36, $p = < 0.001$). Decomposition of wood was increased in warming treatments compared to the control (effect size = 0.17, 95% Confidence Intervals = 0.13, 0.20, $p = < 0.001$). However, removal of warming from the background of the multiple global change factors did not affect the decomposition rate.

The only response variable considering soil structure and properties affected by warming was WSA. WSA increased in warming treatments (effect size = 7.75, 95% Confidence Intervals = 3.39, 13.35, $p = 0.0205$), but removing the factor warming in the multiple global change factor treatments did not have a significant effect.

Drought had mostly neutral effects as single factor and positive effects when it was removed from the factor background. Drought as single factors did not affect enzyme activity, but in the background of the multiple factors, removal of drought increased β-D-glucosidase activity enzyme activity compared to treatments with all factors (effect size = 0.68, 95% Confidence Intervals = 0.31, 1.08, $p = < 0.00671$) as well as acid phosphatase activity (effect size = 2.14, Confidence Intervals = 1.18, 2.88, $p = < 0.001$) and N-acetyl-β-glucosaminidase activity (effect size = 0.43, 95% Confidence Intervals = 0.30, 0.55, $p = < 0.001$). Decomposition of wood was increased in drought treatments compared to the control (effect size = 0.35, 95% Confidence Intervals = 0.25, 0.40, $p = < 0.001$) (Fig. 1d, h). Removing drought from the background of the multiple factors showed a positive effect on decomposition (effect size = 0.09, 95% Confidence Intervals = −0.07, 0.15, $p = < 0.001$). Looking at soil structure and properties, the MWD was negatively affected by the single factor addition of drought (effect size = −0.50, 95% Confidence Intervals = −0.61, −0.33, $p = < 0.001$). The

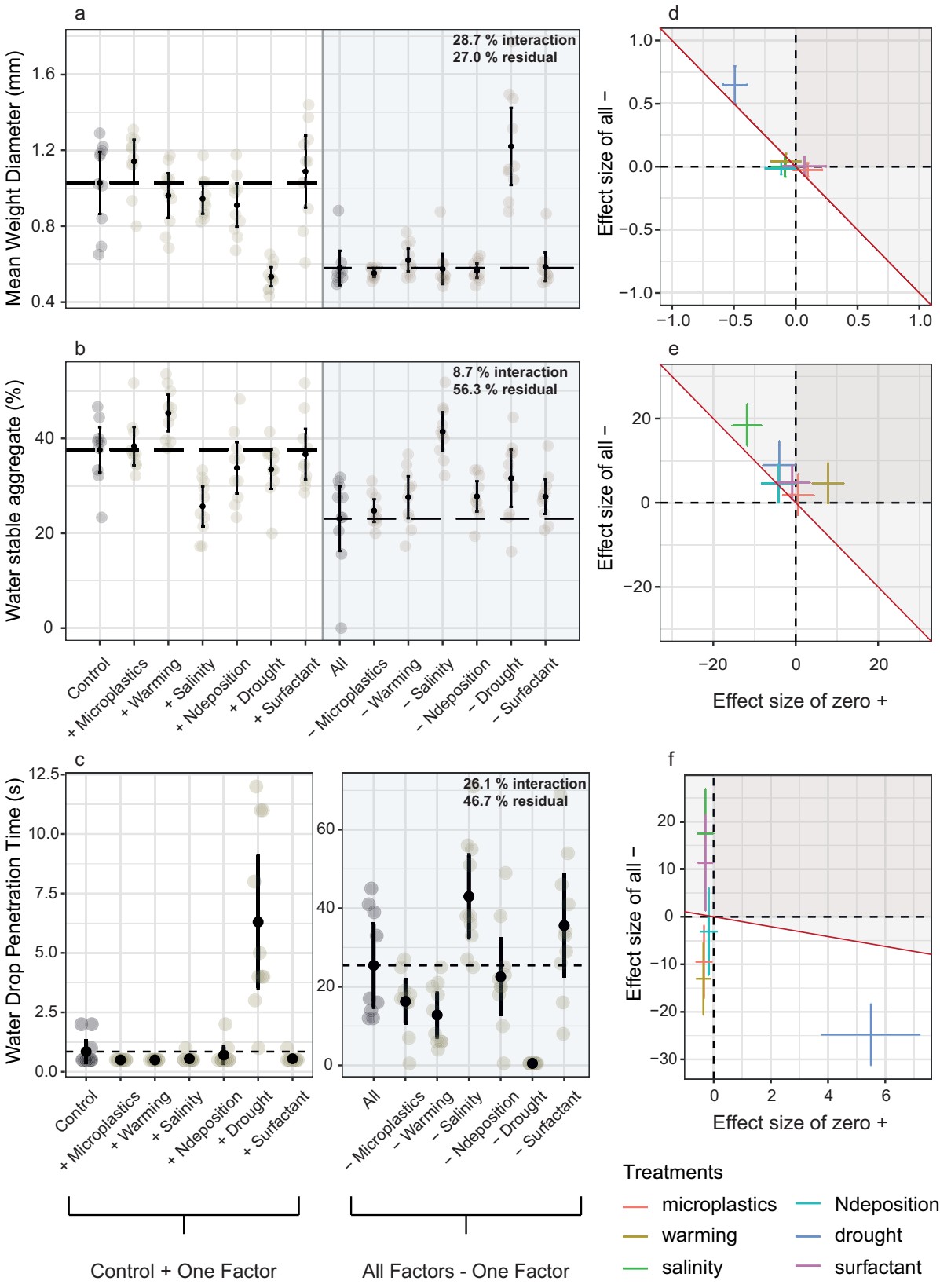

removal of drought from the multiple factor background led to an increase in MWD (effect size = 0.64, 95% Confidence Intervals = 0.47, 0.84, $p = <0.001$).

Drought increased the water drop penetration time (effect size = 5.45, Confidence Intervals = 3.25, 7.80, $p = <0.001$). When drought was removed from the global change factor background there was no

difference between the other global change factors combined and the control.

## Discussion

Using a subtractive experimental design, we here compare the effects of a single global change factor with the effects of that factor acting in

**Fig. 2 | Mean weight diameter (MWD), water stable aggregates (WSA) and water drop penetration time affected by multiple factors of global change.** **a**–**d** The left part of the graph (white background) shows the control and the single factor treatments. The dashed line shows the mean value of the control plots. Dots reflect the measured values (10 replicates each), marked with a black filled dot is the mean, the 95 % confidence interval is shown by the line passing through the dot. On the right side of the graph (gray background), the combination of all global change factors (dashed line as mean), is compared to the treatment combinations, from which each of the single factors is subtracted. Redundancy analysis (RDA) and variance partitioning were performed to quantify the relative contribution of each factor, as well as their interactions, and residual (biological noise or measurement errors), presented in the upper right corner of the (**e**–**h**), To visualize the effect directions of factor addition and subtraction treatments, bootstrapped ($n = 100$) effect size of factor addition were plotted on the $x$ axis and the effect of factor removal on the $y$ axis. Different colors represent different global change factors. Factor effects on the diagonal red line indicate that the effect of the factor in the multi-factor background has the direction and intensity expected from the single-factor effect. Factor effects in the gray shaded area indicate that a factor in combination with other factors has a more negative effect than expected from the single factor effect. Dark gray areas (upper right) represent positive single factor effects that have a negative effect when combined with other factors (effect size of single factor positive, effect size of factor removal positive as well). Source data are provided as a Source Data file.

concert with an assemblage of multiple other factors. Previous experimental designs did not offer the opportunity for this analysis. We find that the effect sizes of factor addition and removal were not always the same, nor did the effect go in the opposite direction in every case; meaning the same factor had opposite effects when acting alone or in concert with others. For example, we observed that even a factor with a positive single factor effect, like warming, can be a burden when jointly acting with multiple global change factors, as seen by its removal from the set of factors eliciting a positive effect, rather than the expected negative effect. We excluded effects via other parameters such as pH and soil nutrients.

## Single factor effects

Supporting our first hypothesis, the chosen global change factors showed positive, negative and neutral single factor effects. Within the single factor treatments, we found negative effects of salinity on phosphatase and glucosidase activity, on decomposition rate and water stable aggregates. Salinity could reduce enzyme activity in several ways: osmotic desiccation of microbial cells releasing intracellular enzymes, altering of the ionic conformation of the active site of the enzyme protein, or a specific ion toxicity causing nutrient imbalance for microbial growth and subsequent enzyme synthesis[15]. Negative effects of salinity on enzyme activity and decomposition are therefore plausible and have also been confirmed by other studies[16,17]. For the water stability of soil aggregates, salinity has been found to increase[16] or decrease WSA[18]. The soil texture and intensity of salt stress may lead to divergence of results. A reduction of WSA in saline soil could be explained by its disintegration properties causing slaking of aggregates, but also by a reduction of activity of microbes involved in aggregate formation and stabilization, such as mycorrhizal fungi[19], or potentially other fungi important for soil aggregation[20,21]. Since the control soil absorbed the water within one second, single factor treatment effects that may have lowered water drop repellency were not detectable. Only in multiple factor backgrounds were effects of anthropogenic factors detectable for this variable. Multiple global change factor treatments very strongly increased water drop penetration time, especially in treatments without salt, suggesting that in a more water repellent control soil, this factor may have had a positive influence. A positive effect of salinity on water drop penetration time has been found in an urban context[22]. However, in an agricultural system, including plants in the field, effects on watering with saline sewage showed the opposite effect of increasing soil water repellency[23].

Warming frequently caused positive effects in the single factor treatments, mainly in N-acetyl-glucosaminidase activity, decomposition, and WSA. A trend of increased N-acetyl-glucosaminidase activity in warming treatments was also found in a meta-analysis on the effects of global change on soil extracellular enzyme activities[24]. Warmed environments without water limitation are thought to promote plant growth and microbial activity[25], increase fungal hyphal growth and enhance secretion of binding agents[26]. This may also lead to higher

water stability of aggregates. However, effects are not always clear. Rillig et al.[27] observed a negative effect of temperature increase on water stability of aggregates in annual grassland.

In our experiment, drought had positive, negative, or neutral effects depending on the response variable. Decomposition was positively affected by drought, MWD negatively. A lower MWD under drought in the upper soil layer due to the loss of macroaggregates was also found in other studies[28]. In our case, however, no change in the supply of organic plant material could have taken place and the enzyme activity of the microorganisms also showed no change under drought stress. Lignin and therefore wood decomposition is mainly performed by fungi, which are more drought tolerant than bacteria[29], but usually, drought decreases wood decomposition[30,31]. Drought also increased the water drop penetration time. However, WSA was not affected. Precipitation likely plays a key role in root-mediated aggregate stability[32], which could be the reason why we did not find effects of drought on WSA, since our experiment did not include plants.

Microplastics, surfactant and N deposition did not have significant effects when applied in isolation. For ß-glucosidase activity, a positive trend of microplastic and surfactant was detected, probably due to the use of these additives as a carbon source[33]. Only ß-glucosidase activity was slightly increased, which supports the hypothesis of Leifheit et al.[33] that tire abrasion could be used as an additional carbon source only by certain microbes, thus leading to shifts in the community. The chemical surfactant sodium dodecyl sulfate can also be rapidly mineralized by microbes, possibly leading to the substance being degraded after 6 weeks of incubation[34]. Positive effects of N addition on enzyme activity were found in a meta-analysis[24] but not in our study. Effects on decomposition or soil structure were also not found, possibly because the dose we used was quite low.

## Combined effects of all factors

As hypothesized (hypothesis 2), higher order factor combinations negatively affected soil functions and processes irrespective of the sign and magnitude of the single global change factor treatments, supporting our second hypothesis and previous work[8]. Environmental changes that lead to new combinations of change factors, such as land use change, often lead to relatively homogeneous effect directions of various response variables, even if some of the individual factors show heterogeneous effect directions[35]. However, experiments combining more than two global change factors are rare[8] and not only factor number, but also their identity and dissimilarity play a role in predicting multiple global change factor effects[9].

In a study testing global change factor effects on a plant community mesocosms a single factor (eutrophication) had a strong effect on plant biomass that was not increased by a combination of six global change factors including eutrophication, while for other response variables, the effect of the combination of factors was stronger than the single factor effects[12]. Similarly, in our experiment, MWD and water drop penetration time did not show a significantly different effect of all factors combined compared to the single factor effect of drought.

Conversely, all multiple global change treatments without drought had a similar aggregate size distribution and water drop penetration time as the control.

## Effects of omitting factors

According to our third hypothesis, effect size magnitudes and directions would differ depending on whether a factor was added as a single factor or removed from a background of multiple global change factors because of synergistic and antagonistic effects of factor combinations. These results can be viewed in two different ways: by observing the effect of the removal of a factor we can deduce how this factor must have acted in concert with all the other factors. In a more indirect way, we can interpret such removal effects also in a more applied sense, namely as the potential benefit of excluding factors in a management context. In our discussion here we initially focus on the first case.

We found positive and neutral effects of removing factors, but no negative effects, meaning all factors contributed to negative effects in the context of the set of factors present. Removing salinity from a background of multiple global change factors was beneficial for all measured response variables, except for MWD, where single factor addition also did not cause an effect. Warming, despite many positive single factor effects, had negative effects in a background of other global change factors. One mechanism of warming directly influencing other factors of global change is shifting reaction rates[36]. Potentially, warming increased the concentration of salts in the soil solution, since water has evaporated due to increased temperature and therefor acts as a concentration amplifier[36]. Drought as well leads to a decrease in soil water content and therefore other factors become more concentrated[36]. Consistent with this, Qadeer et al.[37] found negative effects of the single factors drought and salinity on soil extracellular enzyme activity and stronger negative effects of the combination of these stressors. In our experiment global change factor combinations without drought but with salinity show a higher decomposition rate than when both treatments are combined most likely because an increased water availability mitigates the osmotic effects of salinity.

The trend of increased ß-glucosidase in the single treatments of microplastics and nitrogen addition could not be detected when comparing the removal of these factors with the combined factor treatment. Given the potential use of these substances as a resource for microbes and the reduction in overall enzyme activity in soil treated with multiple global change factors, the already small single factor effects of microplastics and nitrogen were not detectable. A lower abundance or activity of microbes could be an explanation that would reduce the effect of adding a resource. Using maize residues as a resource Wichern, Wichern and Joergensen[38] found a positive effect on microbial biomass in soils with low salinity, but no effect in soils with high salinity. This supports the notion that a multi-factor background that already limits microbial processes can cause factors that represent a provision of resources to have limited effects. We found a higher proportion of unexplained variance in enzymatic activity, which could be due to measurements being taken in small volumes, for example. As an indicator of soil health, enzymatic activity is recommended for use alongside other biological or physicochemical measurements, as we do here[39].

Compared to studies that test gradients of an increasing number of global change factors[8,9,11,12,40], the subtractive study design also provides information on the effect of a single factor in combination with all other factors; this is not possible in the 'additive' designs used previously, since the identity of factors added along the factor number gradient created in such experiments is by necessity randomly determined. While we now know how factors contribute to overall effects in a high-dimensional background, the mechanisms underlying such effects need to still be elucidated. It is likely that not only direct effects of factor interactions influenced our results, but also indirect effects of factors on soil microorganisms, leading to non-additivity of factor combinations[36]. Future experiments should include more complex systems including other groups of organisms, specifically plant-soil systems.

Novel experimental designs are urgently needed to learn more about the mechanisms of how multiple factors affect soils and ecosystems. The subtractive design we introduce here can provide insights into the different effects of factors by omitting them from a set of jointly acting factors. This design can be leveraged specifically to learn about the high-dimensionality end of factor interactions, by directly manipulating factor presence in the background of larger groups of interacting factors. We conclude that the effect size and direction of a factor added in isolation does not carry information about the effect of removing this same factor from a set of interacting factors - not even in terms of effect direction. Our results are potentially highly relevant for the restoration of ecosystems (even though we did not remove factors in our experiment, as would be the case in restoration practice, but rather did not add them). Based on our results, it would be advisable to first target those factors that cause the strongest negative effects in multiple-factor situations, not necessarily the factors causing the strongest negative single effects; in our case for example salinity was the factor with the strongest negative single factor effect and the strongest negative effect in multiple-factor settings. However, warming had mostly positive single factor effects but strong negative effects when interacting with multiple global change factors and mitigating this factor may thus be equally important. Our study can serve as a blueprint for additional work testing these ideas in other contexts and at higher levels of system complexity (e.g., with plant communities). It can as well be applied in different contexts, where factor gradients have been tested, to ask a different question: does the effect of a single factor differ when tested in isolation from the effect it has in a multi-factor context - irrespective of whether it is multiple factors of global change[8–12], restoration amendments[41,42], or species[43,44].

## Methods
### Experimental set up

We used 140 50 mL tubes (Corning propylene centrifuge tubes) as experimental units. Screw-top caps had holes with hepa filters to allow ventilation while preventing microbial contamination. Temperature sensors were integrated into two additional tubes. The tubes were placed in cups filled with sand to stabilize and thermally isolate them from neighboring units. Tubes were filled with 25.0 g of soil (from a grassland experimental site of Freie Universität Berlin (52° 33′ 09.53″N, 12° 40′ 07.86″ E), an Albic Luvisol containing 6% sand, 18.8% silt and 7.6% clay; 6.9 mg/100 g P; 5.0 mg/100 g K (analyses conducted by LUFA Rostock Agricultural Analysis and Research Institute, Germany). Soil was dried at room temperature and sieved to 2 mm). 5.0 g of soil was sterilized (autoclaved, 121 °C, 1 h) to be used as "loading soil": this soil was mixed with the appropriate dose of each treatment and then added to the 25.0 g of soil to be filled in the conical tubes. We used this loading soil to achieve more effective mixing of the treatments; the soil was sterilized to avoid exaggerated effects on the soil community due to the higher concentration of chemicals added to this soil. The experiment was set up in a climate chamber (incubation in the dark, 60% relative humidity, ambient temperature, relative humidity, ambient temperature 20 °C. At the start of the experiment, we added water (equivalent to 60% water holding capacity) to each experimental unit (except for treatments containing drought, which received an amount of water equivalent to 30% water holding capacity). After three weeks, the water holding capacity was readjusted. The experiment ran for six weeks.

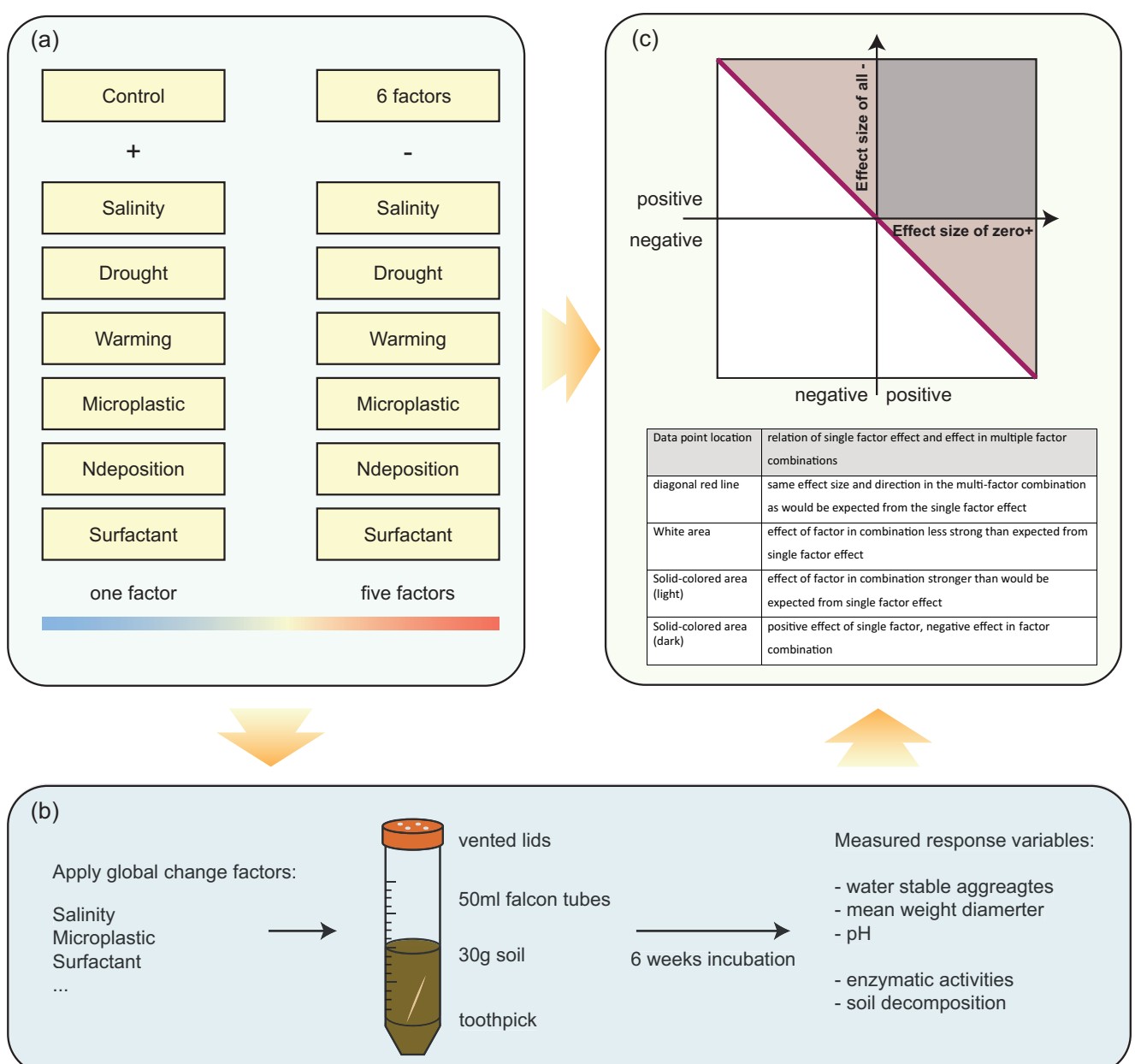

**Fig. 3 | Study design and possible outcomes. a** Treatments were warming, drought, salinity, N deposition, microplastic (tire wear particles), and surfactant. Ten untreated replicates were used as control. Each single factor treatment (+), the combination of all global change factors, and treatments in which one of the factors was dropped (-) were each replicated ten times. The experiment consisted of 140 experimental units in total. **b** Experimental units were falcon tubes containing 30 g of soil. A pre-weighed toothpick was included for measurement of decomposition rate. After six weeks of incubation, response variables were measured. **c** To visualize the effect directions of factor addition and subtraction treatments, we show the bootstrapped ($n = 100$) effect size of factor addition on the $x$ axis and the effect of factor removal on the $y$ axis.

## Experimental design

Treatments were the global change factors warming, drought, salinity, N deposition, microplastic (car tire abrasion), and surfactant in doses that were detected in urban environments (see below and Fig. 3). The control was compared with the single factor treatment addition (+) and the combination of all global change factors was compared with treatments in which one of the factors was dropped from the background of the multiple global change factors (-). Each treatment was replicated ten times. The temperature treatment was applied using a heating cable wrapped around the individual tubes (PT2011, Exo Terra, Germany) with a separate controller per experimental unit (ETC-902, VOLTCRAFT, Germany). Temperature sensors were installed to monitor the heating cable

performance, logged temperature mean was 20.54 °C (sd 0.4) for control and 26.44 °C (sd 0.49) for the warming treatment. Warming was achieved by increasing the temperature by 5 °C, as can be observed in centers of highly urbanized cities[45], and as is predicted for long-term climate change. Drought was 30% of water holding capacity compared to 60%[46,47]. Salinity was achieved by adding NaCl until a moderate electrical conductivity of 3.0 dS m-1 was reached (Supplementary Fig. 2c)[48]. As nitrogen deposition treatment, $NH_4NO_3$ was added to the equivalent of 18 kg N ha −1 yr −1 to simulate urban N deposition[49]. Tire wear particles were added at a concentration of 1 g per kg soil[33] and the surfactant sodium dodecyl sulfate was added at 16 mg per kg soil. Surfactant concentrations in graywater effluents can range from 0.7 to 70 mg L[−1 50], and from 0.2

to 20 g kg⁻¹ in sewage sludges, leading to soil concentrations of up to several mg kg⁻¹[51]. Experimental studies involving surfactant application have used different dose levels[52]. A synthetic anionic surfactant was applied at concentrations of 16 mg kg⁻¹ in both a field study (Figge & Schöberl, cited in ref. [53]) and a greenhouse experiment[9]. Based on these, we selected this concentration of sodium dodecylbenzenesulfonate to simulate a surfactant-contaminated hotspot.

## Response variables

We measured enzyme activity (β-D-glucosidase, acid phosphatase, and N-acetyl-β-glucosaminidase), decomposition, mean weight diameter of soil aggregates (MWD), percentage of water stable aggregates (WSA) and water drop repellency. The measurements were chosen to represent interconnected processes related to cycling of matter and structuring of soil. We tested electrical conductivity and pH, as well as C, N and P content to support our interpretation and exclude effects not caused by the treatments (Supplementary Figs. 2 and 3).

**Enzyme activity.** β-D-glucosidase (EC3.2.1.21), acid phosphatase (EC3.1.3.2), and N-acetyl-β-glucosaminidase (EC3.2.1.52) were measured from 5 g of soil. Water content for drought treatments was adjusted. The method of using high throughput microplates assays was described by Jackson, Tyler and Millar[54]. Briefly, 5 g soil was mixed with 10 ml 50 mM acetate buffer (pH 5.0–5.4) in a 50-ml falcon tube. Then, 150 ul of soil slurry for each sample was vortexed and pipetted into six wells on a 96-deep well plate. 150 ul substrate solutions (5 mM 4-$p$-nitrophenyl-β-glucopyranoside5 mM, 4-$p$-nitrophenyl-phosphate disodium salt hexahydrate, and 2 mM 4-$p$-nitrophenyl-β-N-acetylglucosaminide, Sigma, Germany, item no.: N71768, N7006, N5759, and N9376) were added to the first four wells, 150 ul acetate buffer was added into the last two wells of each sample (buffer control). Plates were incubated in the dark at 25 °C for 2 h (for β-D-glucosidase and acid phosphatase) or 4 h (N-acetyl-β-glucosaminidase). After the incubation, plates were centrifuged at 2000 x g for 5 min, and then 100 ul supernatant from each well was transferred into new microplates containing 10 ul 1 M NaOH and 190 ul distilled water in each well. Finally, the absorbance at 410 nm was recorded by a microplate reader (Benchmark Plus Microplate Spectrophotometer System, BioRad Laboratories, Hercules, CA, United States).

**Decomposition.** We inserted a pre-weighed wooden toothpick into the soil, which was retrieved at the end of the experiment. The weight of the dried toothpicks after six weeks of incubation in the soil was used as an indication of decomposition (percent decomposition).

**Mean weight diameter (MWD).** Soil was dried at 60 °C and sieved (4 mm). Then we measured the aggregate size distribution and stability by following the protocol by Kemper and Rosenau[55]. The dry sieving technique was performed in a modified way, using a sieve cascade consisting of sieves with mesh sizes of 2 mm, 1 mm, 250 μm, 100 μm and 53 μm. A collecting tray was located under the sieve stack. By gentle horizontal shaking, the aggregates were separated into different size fractions. Mean weight diameter was calculated according to the following formula:

$$MWD = \sum_{i=1}^{7} \frac{w_i}{w_{sample}} x d_i' \qquad (1)$$

where $i$ is the fraction size, $W_i$ is the weight of each fraction, $d_i$ is the mean diameter of each size fraction. This commonly used index weighs more heavily larger sized aggregate fractions.

**Water stable aggregates (WSA).** We used a wet sieving technique to determine the percentage of water-stable aggregates, modified after Kemper and Rosenau[55]. Sieves (mesh size 250 μm, for measuring macroaggregates) were filled with 4.0 g of dried soil. The soil samples were capillarily rewetted and then placed in a wet sieving machine (Agrisearch Equipment, Eijkelkamp, Giesbeek, Netherlands). The samples were moved up and down in a water column for 3 minutes. The soil was then dried at 60 °C. After drying, the weight of the dry matter was recorded. The soil was then transferred back to the sieve and passed through to obtain the coarse matter fraction.

Water-stable aggregate percentage was calculated as:%WSA = (water stable aggregates − coarse matter)/(4.0 g − coarse matter)

**Soil water repellency.** The water drop penetration time was measured, where three droplets (8 μl) of deionized water are placed on the soil surface, and the time in seconds is counted until each droplet soaks in ref. [56].

## Statistical analysis

Statistical analysis was conducted in R 4.2.1[57]. 'dabestr'[58], 'Rmisc'[59], vegan'[60], and 'tidyverse'[61] were used for data analysis, 'ggplot2'[62] and 'pheatmap'[63] were used for plotting the figures. Effect sizes were normalized with standard deviation normalization (z-score) method and calculated by bootstrapping ($n = 100$). Anova and Tukey's test were used to test for statistical significance of treatment effects on the tested response variables.

To quantify the relative contribution of each factor and their interactions for the all- group, and ascertain if biological noise or measurements error could be responsible for some of the patterns we observed, we performed redundancy analysis (RDA) using the vegan package. A higher $R^2$ value represents stronger explanatory power, reflecting a more robust effect.

In the RDA model, the total variance of each soil response is partitioned into three components: variance explained by the independent global change factors, the shared (interactive) variance that is jointly explained by multiple global change factors, and the residual variance, representing the proportion of total variance that remains unexplained in the model.

For the variance explained by the independent global change factors, the 6 factors were used as explanatory variables and the adjusted coefficient of determination was obtained using RsquareAdj(). We subsequently removed one factor i from the model and refitted the RDA model (Formula 1), where $\Delta R_i^2$ represents the variance explained by factor i.

$$\Delta R_i^2 = R_{6factor}^2 - R_{removingi}^2 \qquad (2)$$

The shared (interactive) variance, that indicates interaction across factors of global change, was calculated with Formula 2, indicating how much variance cannot be attributed to any single factor and is shared among factors.

$$R_{shared}^2 = R_{6factor}^2 - \sum_{i=1}^{6} \Delta R_i^2 \qquad (3)$$

The residual variance indicates the variance that the model cannot explain, calculated as $1 \cdot R_{6factor}^2$, and likely reflects biological noise or measurement error. We included the variance explained by single factors, shared variance and residual for each soil responses in the supplementary information.

## Reporting summary

Further information on research design is available in the Nature Portfolio Reporting Summary linked to this article.

# Data availability

The experimental data generated in this study have been deposited in the Figshare database with DOI: 10.6084/m9.figshare.28151276

(https://doi.org/10.6084/m9.figshare.28151276). Source data are provided with this paper.

## Code availability

The code for reproducing all statistical analyses in this manuscript is published in the Figshare database with (https://doi.org/10.6084/m9.figshare.28152140).

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

## Acknowledgements

H.L. acknowledges the China Scholarship Council for a scholarship (202108080156).

## Author contributions
RR designed the research and wrote the first draft, HL designed the figures and performed the statistical analysis, MCR and AL contributed to interpreting the results, editing of manuscript, and the drafting of the figures. RR, HL, AL, and AW collected the data.

## Funding

## Competing interests
The authors declare no competing interests.
