## [Transparent Peer Review file · Nature Communications]

Global change factors differ in effect when acting alone and in a multi-factor background

Corresponding Author: Professor Matthias Rillig

A version of this paper was originally rejected for publication by Nature Communications, however that decision was reconsidered after appeal by the authors.

Version 0:

Reviewer comments:

Reviewer #1

(Remarks to the Author)

"Rebecca Rongstock et al. present results regarding the response of urban soil to multiple environmental factors, including warming, drought, salinity changes, nitrogen deposition, tire abrasion, and surfactants. While the paper is technically well-documented, it lacks both mechanistic interpretation and a demonstration of broad impact. More importantly, similar work has been published elsewhere [Meidl, P., Lammel, D. R., Nikolic, V., Decker, M., Bi, M., Hampl, L., & Rillig, M. C. (2024). Combined Application of Multiple Global Change Factors Negatively Influences Key Soil Processes across an Urban Gradient in Berlin, Germany. *Soil Systems*, 8(3), 96.]

Existing literature extensively explores soil processes (biogeophysical and biogeochemical) in response to multifactor environmental changes. This work, while providing site-specific evidence, does not clearly articulate its novel scientific contribution to the field at a high-impact level. The following points offer specific feedback for consideration:

1. Lack of a clear research question

The abstract, introduction, and even the results section fail to define a distinct scientific question that the study aims to address. For instance, what are the most relevant questions that could be explored using this multifactor experimental design? Why are these questions significant at a high-impact level? What knowledge gaps in the existing literature hinder our ability to answer these questions, and how does this study innovatively address them?

2. Insufficient mechanistic interpretation

The results section primarily reads as a descriptive account of experimental outcomes, lacking a synthesis of higher-level knowledge. The analysis does not lead to significant implications, nor does it offer a hypothesis-driven discussion of the observed results. Consequently, mechanistic interpretations are largely absent.

3. While the subtractive experimental design is acknowledged, the paper does not adequately justify its superiority over other perturbation designs. In reality, environmental perturbations are neither strictly additive nor subtractive. Instead, multiple change factors interact simultaneously in a complex way, potentially exerting differential impacts on urban soils across different spatial and temporal scales.

4. Limited broad Impact and methodological Focus:

The paper's overall impact is limited, as it primarily functions as a methodology-oriented technical report. A similar study, potentially from the same research group, has been published (Meidl, P., Lammel, D. R., Nikolic, V., Decker, M., Bi, M., Hampl, L., & Rillig, M. C. (2024). Combined Application of Multiple Global Change Factors Negatively Influences Key Soil Processes across an Urban Gradient in Berlin, Germany. *Soil Systems*, 8(3), 96.).

5. In summary, this paper does not appear to meet the high standards and scientific impact expected for publications in journals such as Nature Communications."

Reviewer #2

(Remarks to the Author)

Rongstock and co-authors studied how multiple global change factors interact to impact soil functioning and parameters. The authors rather to increase the number of factors combined, used a subtractive experimental design, where each factor effect was study alone as well as their importance when being removed from the all combined factor treatment. This type of design is highly novel and help to better understand how a specific factor may influence the direction of the impact when being part of a multifactor treatment. Multifactorial experiments have often the limitation to not have enough replicates while covering all interactions. Here, the authors demonstrate in an elegant way how a multifactorial experiment can be done to answer a question without studying all interactions.

The authors showed that the interactive effects of multiple factors differed from their main effects, suggesting that mitigation efforts should focus on the most detrimental factors in multi-stressor environments rather than those with significant main effects. I have only minor comments that after being addressed, will improve the readability of the manuscript.

L199. Add a coma after experimental design.

L221. Not only mycorrhizal fungi participate to soil aggregation. I would rephrase it to englobe all fungi producing hyphae.

L226. Add a comma after soil.

Figures:

Figure 2-3 I would suggest the authors to decrease transparency of the color for each treatment and/or use more contrasting colors. The shade of grey used in the effect size panels make is difficult to distinguish colors and thus the different treatments.

L392 Space between N and deposition missing.

Figure 2. The unit for the decomposition rate is missing.

Supplementary material: add in the table caption, the meaning of control 0 and control all.

Version 1:

Reviewer comments:

Reviewer #2

(Remarks to the Author)

Dear Authors,

I think you did a great job taking into account my comments. According to the version of the supplementary material that I could download on the platform, the added sentence explaining the difference between control_0 and control_all was not added, please do so.

I was also asked to review your response to reviewer 1. I think the updated version of the manuscript indeed highlight better the novelty of the study and makes the authors'work more readable.

I have no further comments.

Reviewer #3

(Remarks to the Author)

While the study employs an innovative subtractive experimental design to explore the interactions of multiple global change factors (warming, drought, salinity, nitrogen deposition, tire abrasion, and surfactants) on urban soil processes, a significant concern that prevent me from recommending publication in its current form. The paper's methodological limitations, lack of mechanistic depth, and overstated implications undermine its scientific contribution.

1. The authors emphasize the novelty of their subtractive design as a key innovation. However, this approach bears strong resemblance to prior factorial and gradient-based studies in multifactorial ecology (e.g., Meidl et al., 2024, cited in the manuscript, which also examines urban soil responses to multiple stressors). Additionally, the subtractive method is analogous to omission designs used in agricultural and restoration studies (e.g., Li et al., 2025, cited in references), where factors are systematically excluded to assess their contributions. The manuscript fails to convincingly demonstrate how this design provides a quantum leap beyond existing frameworks, such as factor-number gradients (e.g., Rillig et al., 2019), rather than an incremental adjustment. Without a clear articulation of unique theoretical or empirical advances, the work reads as a technical refinement rather than a high-impact contribution.

2. The experimental setup (using 50 mL tubes with sterilized "loading soil" in a climate chamber) severely limits the ecological relevance and generalizability of the findings. Urban soils are heterogeneous systems influenced by dynamic

interactions with plants, fauna, and spatial-temporal gradients (as acknowledged in the introduction but not addressed). This artificiality raises questions about whether the observed factor interactions, such as salinity's negative effects or warming's context-dependent impacts, which would manifest similarly in natural urban environments.

3. The results section is predominantly descriptive, cataloging effect sizes without delving into underlying biological or physicochemical mechanisms. For instance, the manuscript reports that warming had positive single-factor effects but negative impacts in multifactorial contexts, attributing this to potential "concentration of salts" due to evaporation. Yet, this is speculative and lacks supporting data on microbial community shifts, enzyme kinetics, or soil physicochemical changes (e.g., ion concentrations or organic matter dynamics).

4. The statistical approach relies heavily on bootstrapped effect sizes and ANOVA with Tukey's test, which may not adequately capture the complexity of high-dimensional interactions. The authors report "effect sizes not always the same" but provide no quantification of interaction strengths or variance partitioning (e.g., how much variability is explained by individual factors vs. higher-order interactions). This is critical given the mixed effects (e.g., drought increasing decomposition but decreasing MWD), which could stem from experimental noise rather than biological significance.

5. Surfactant concentrations ("16 mg per kg soil") are not justified against real-world levels.

6. In summary, the manuscript addresses an important topic in urban soil ecology but falls short of the high standards expected for publication.

Point-by point response to Reviewers' comments:

Reviewer #1 (Remarks to the Author):

"Rebecca Rongstock et al. present results regarding the response of urban soil to multiple environmental factors, including warming, drought, salinity changes, nitrogen deposition, tire abrasion, and surfactants. While the paper is technically well-documented, it lacks both mechanistic interpretation and a demonstration of broad impact. More importantly, similar work has been published elsewhere [Meidl, P., Lammel, D. R., Nikolic, V., Decker, M., Bi, M., Hampl, L., & Rillig, M. C. (2024). Combined Application of Multiple Global Change Factors Negatively Influences Key Soil Processes across an Urban Gradient in Berlin, Germany. *Soil Systems*, 8(3), 96.]

Existing literature extensively explores soil processes (biogeophysical and biogeochemical) in response to multifactor environmental changes. This work, while providing site-specific evidence, does not clearly articulate its novel scientific contribution to the field at a high-impact level. The following points offer specific feedback for consideration:

>> Thank you for your critical and constructive appraisal of our work. The paper published before (Meidl et al.) is in fact not similar to the work presented here (as it uses a completely different experimental design); the reviewer likely reached this conclusion because of a lack of clarity in presenting the novelty of our approach in the manuscript. This has now been rectified. We found your comments to be exceptionally helpful in improving the presentation and increasing the potential impact of the work. We summarize responses to individual points in your review below.

1. Lack of a clear research question

The abstract, introduction, and even the results section fail to define a distinct scientific question that the study aims to address. For instance, what are the most relevant questions that could be explored using this multifactor experimental design? Why are these questions significant at a high-impact level? What knowledge gaps in the existing literature hinder our ability to answer these questions, and how does this study innovatively address them?

>>>> Thank you for pointing out that some explanations lack clarity; we agree. We rewrote parts of the abstract and introduction, to make the research question clearer. In short, we ask if factor effects differ in their effect size and direction when comparing the addition of single factors to an untreated control vs. when a factor is subtracted from a set of multiple other co-occurring factors. Knowing which individual factors have particularly strong effects in multifactorial contexts improves our ability to decide which factors should be reduced or eliminated first in any restoration efforts. **In addition, this is the first work using this systematic subtractive experimental approach in a high-factor background, and as such it is clearly distinguished from any previous experimental approaches in ecology and the environmental sciences. Our work introduces an innovative template of an experimental design that can now be applied widely in the environmental sciences and beyond.**

2. Insufficient mechanistic interpretation

The results section primarily reads as a descriptive account of experimental outcomes, lacking a synthesis of higher-level knowledge. The analysis does not lead to significant implications, nor does it offer a hypothesis-driven discussion of the observed results. Consequently, mechanistic interpretations are largely absent.

>>>> Thank you for this comment. We agree with the reviewer and acknowledge that the Results section did not contain many statements that facilitate a high-level synthesis. We have alleviated this in the revision. For example, the Results section now starts with this new lead-in paragraph, emphasizing the novelty of what has been accomplished: *"We here successfully pioneered an experimental approach to examine the effects of factors in a high-dimensionality factor background by using a subtractive design, where systematically one factor is removed from the set of the other factors. In comparing this subtractive design with adding each factor by itself to the control, to measure its effect in isolation, we find that the effects of a factor acting alone can be opposite in direction to the effect when this factor acts in the context of all the other factors."*

We have also changed all section headings from the descriptive "Effects of..." to a statement encapsulating the main result to help guide the reader to main findings.

Throughout, we have introduced wording changes to make clearer what is new and what confirms previous findings. For example, we add a lead-in to the key section about lack of correspondence of single and joint effects: *"We reveal that a factor acting by itself has effects that diverge from the effects of that factor co-acting with other factors, even to the point that the direction of the effect changes."*

3. While the subtractive experimental design is acknowledged, the paper does not adequately justify its superiority over other perturbation designs. In reality, environmental perturbations are neither strictly additive nor subtractive. Instead, multiple change factors interact simultaneously in a complex way, potentially exerting differential impacts on urban soils across different spatial and temporal scales.

>>>> Thanks for this comment. We added a paragraph in the introduction to describe the benefits of the subtractive design in comparison to the factor gradient approach: *"Other studies using factor gradients have drawn similar conclusions for different plant and soil systems^{9–12}. While factor gradients can be applied to ask whether the effect of multiple interacting factors can be predicted by single factor effects, we propose here a novel design that provides a step toward a more mechanistic understanding, focusing on the effect of individual factors in isolation and within a multifactorial context with implications for restoration efforts."*

And in the conclusion, we added that the design is more widely applicable: *"The design we pioneered here for the examination of factor effects in a multi-factor background of high dimensionality can be readily applied in different contexts where asking questions about factor numbers is relevant: does the effect of a single factor differ when tested in isolation*

from the effect it has in a multi-factor context - irrespective of whether it is multiple factors of global change 8–12, restoration amendments 40,41, or species 42,43.”

The complexity of global change factor interaction was reduced in the experiment. In reality, there are of course more factors, with heterogeneity in distribution and gradients, rather than treatment vs. control. However, **compared to studying single factors, or factor pairs, constituting over 98% of all published papers on this topic** (Rillig et al. 2019, Science), it is a big step towards a much more realistic experimental setting. We have added this point to the first section of the Conclusions, which now reads: *“Novel experimental designs are urgently needed to learn more about the mechanisms of how multiple factors affect soils and ecosystems, given that 98% of all papers in global change and soil deal with only one or two factors 8.”*

In addition, the main goal of this experiment was not to represent reality but to mechanistically explore interactions at the very high dimensionality range; something that has not been done before. We have now emphasized this point (see above).

4. Limited broad Impact and methodological Focus:

The paper's overall impact is limited, as it primarily functions as a methodology-oriented technical report. A similar study, potentially from the same research group, has been published (Meidl, P., Lammel, D. R., Nikolic, V., Decker, M., Bi, M., Hampl, L., & Rillig, M. C. (2024). Combined Application of Multiple Global Change Factors Negatively Influences Key Soil Processes across an Urban Gradient in Berlin, Germany. *Soil Systems*, 8(3), 96.).

>>>> Thanks for this comment, which we have taken as an opportunity to clarify the novelty of our manuscript. While there are some superficial similarities between the two studies in terms of the general urban context and the choice of global change factors, the experimental approaches are completely different. The study of Meidl et al. (which is indeed from the same lab) is now cited along with other, similar studies using factor gradient approaches. The difference between that design (first used in previous work such as Rillig et al. 2019 Science) and the one we use in the present manuscript is quite substantial, and is explained in more detail now. **There is no previous study that uses this systematic subtractive approach, which is the heart piece of the present paper.**

5. In summary, this paper does not appear to meet the high standards and scientific impact expected for publications in journals such as Nature Communications."

>>>> We hope that the improvements we could make to the paper thanks to your helpful comments will have changed this perception.

Reviewer #2 (Remarks to the Author):

Rongstock and co-authors studied how multiple global change factors interact to impact soil functioning and parameters. The authors rather to increase the number of factors combined,

used a subtractive experimental design, where each factor effect was study alone as well as their importance when being removed from the all combined factor treatment. This type of design is highly novel and help to better understand how a specific factor may influence the direction of the impact when being part of a multifactor treatment. Multifactorial experiments have often the limitation to not have enough replicates while covering all interactions. Here, the authors demonstrate in an elegant way how a multifactorial experiment can be done to answer a question without studying all interactions.

>>>> Thank you so much for your positive comments, and for recognizing the novelty of our approach to study multiple concurrent factors!

The authors showed that the interactive effects of multiple factors differed from their main effects, suggesting that mitigation efforts should focus on the most detrimental factors in multi-stressor environments rather than those with significant main effects. I have only minor comments that after being addressed, will improve the readability of the manuscript.

L199. Add a coma after experimental design.

>>>> Thank you, we added the comma.

L221. Not only mycorrhizal fungi participate to soil aggregation. I would rephrase it to englobe all fungi producing hyphae.

>>>> Yes, we took the mycorrhizal fungi as an example, because they were studied in the cited literature. We now added another subordinate clause to explicitly include other fungi: "... but also by a reduction of activity of microbes involved in aggregate formation and stabilization, such as mycorrhizal fungi, or potentially other fungi important for soil aggregation" with citations on an observational study using a salinity gradient and one, that tests different strains of fungi affecting soil aggregation.

L226. Add a comma after soil.

>>>> The comma is added.

Figures:

Figure 2-3 I would suggest the authors to decrease transparency of the color for each treatment and/or use more contrasting colors. The shade of grey used in the effect size panels make is difficult to distinguish colors and thus the different treatments.

>>>> Thanks for this suggestion, we decreased transparency of the light area from 15% to 10%, the dark area was changed from 45% to 15%.

L392 Space between N and deposition missing.

>>>> Thanks, spaces between N and deposition are added.

Figure 2. The unit for the decomposition rate is missing.

>>>> Thank you. The unit of decomposition rate is in decimal/ proportion form, we added the unit as (-) in the Figure.

Supplementary material: add in the table caption, the meaning of control 0 and control all.

>>>> Effect size means (mean_ES) and 95 % confidence intervals (CI_low and CI_high being the lower and upper CI boarder) for all added single treatments compared the control without any treatments (control_0) and the subtracted single factors compared to the combination of all factors (control_all).

Second Revision:

Reviewer #2 (Remarks to the Author): #make Huiying second first co-author?

Dear Authors,

I think you did a great job taking into account my comments. According to the version of the supplementary material that I could download on the platform, the added sentence explaining the difference between control_0 and control_all was not added, please do so.

I was also asked to review your response to reviewer 1. I think the updated version of the manuscript indeed highlight better the novelty of the study and makes the authors'work more readable.

I have no further comments.

Thank you! The sentence is now added.

Reviewer #3 (Remarks to the Author):

While the study employs an innovative subtractive experimental design to explore the interactions of multiple global change factors (warming, drought, salinity, nitrogen deposition, tire abrasion, and surfactants) on urban soil processes, a significant concern that prevent me from recommending publication in its current form. The paper's methodological limitations, lack of mechanistic depth, and overstated implications undermine its scientific contribution.

1. The authors emphasize the novelty of their subtractive design as a key innovation. However, this approach bears strong resemblance to prior factorial and gradient-based studies in multifactorial ecology (e.g., Meidl et al., 2024, cited in the manuscript, which also examines urban soil responses to multiple stressors). Additionally, the subtractive method is analogous to omission designs used in agricultural and restoration studies (e.g., Li et al., 2025, cited in references), where factors are systematically excluded to assess their contributions. The manuscript fails to convincingly demonstrate how this design provides a quantum leap beyond existing frameworks, such as factor-number gradients (e.g., Rillig et al., 2019), rather than an incremental adjustment.

Without a clear articulation of unique theoretical or empirical advances, the work reads as a technical refinement rather than a high-impact contribution.

Thank you for your comment. The subtractive design we used here is different from any of the other published works. We added more detail to the explanation of the benefits of this novel design in the introduction. Thereby we tried to make it more clear that factor gradient designs tackle different scientific questions than the subtractive design we employed here for the first time. Our study on “The dissimilarity between multiple management practices drives

the impact on soil properties and functions” (2025) also uses a factor gradient approach, and thus is not a similar design to the one used here.

We also recapped briefly the unique advantages of this subtractive design at the beginning of the Discussion to highlight that the analyses we conducted here were only possible with this design, and in combination with replicating the single- factor effects in isolation.

2. The experimental setup (using 50 mL tubes with sterilized "loading soil" in a climate chamber) severely limits the ecological relevance and generalizability of the findings. Urban soils are heterogeneous systems influenced by dynamic interactions with plants, fauna, and spatial-temporal gradients (as acknowledged in the introduction but not addressed). This artificiality raises questions about whether the observed factor interactions, such as salinity's negative effects or warming's context-dependent impacts, which would manifest similarly in natural urban environments.

Thanks for your comment. For the first test of the design, we chose a more controlled and simple system to reduce confounding variables and to be able to increase causality. Realism is limited in the controlled systems in the climate chamber, our study has this in common with the vast majority of lab experiments. We do not believe this is a limitation of the study per se, which was aimed to be a proof of principle for this new design; but we completely agree with the reviewer’s point on ecological relevance. In the future, we would be very interested in testing more complex systems to investigate a spectrum from highly controlled, to very realistic systems. We added text to the limitations section and proposed future experiments involving more complex test systems.

3. The results section is predominantly descriptive, cataloging effect sizes without delving into underlying biological or physicochemical mechanisms. For instance, the manuscript reports that warming had positive single-factor effects but negative impacts in multifactorial contexts, attributing this to potential "concentration of salts" due to evaporation. Yet, this is speculative and lacks supporting data on microbial community shifts, enzyme kinetics, or soil physicochemical changes (e.g., ion concentrations or organic matter dynamics).

Thanks for raising this point. We now also tested electrical conductivity of our samples and measured pH, as well as C, N and P content to support our interpretation and exclude effects of other mechanisms. All samples were dried and the same amount of VE water was added for the measurement, so that we could not detect effects of lower soil moisture. We instead included a paper on mechanisms underpinning non-additivity in the discussion that supports our interpretation. We added a material and methods section and figures on the new measurements in supplementary information.

4. The statistical approach relies heavily on bootstrapped effect sizes and ANOVA with Tukey's test, which may not adequately capture the complexity of high-dimensional interactions. The authors report "effect sizes not always the same" but provide no

quantification of interaction strengths or variance partitioning (e.g., how much variability is explained by individual factors vs. higher-order interactions). This is critical given the mixed effects (e.g., drought increasing decomposition but decreasing MWD), which could stem from experimental noise rather than biological significance.

Thank you for pointing this out; we agree with this important comment and have performed additional statistical analyses to address it. We conducted a redundancy analysis (RDA) and variance partitioning to quantify the proportion of variation in each soil response explained by individual factors and by their shared interactions. We added our analytical approach in the methods section as follows:

'To quantify the relative contribution of each factor and their interactions for the all- group, we performed redundancy analysis (RDA) using the vegan package. A higher R^2 value represents stronger explanatory power, reflecting a more consistent effect. In the RDA model, the 6 factors were used as explanatory variables and the adjusted coefficient of determination was obtained using $RsquareAdj()$. We subsequently removed one factor i from the model and refitted the RDA model (Formula 1), where ΔR_i^2 represents the variance explained by factor i .

$$\Delta R_i^2 = R_{6factor}^2 - R_{removing\ i}^2$$

The shared (interactive) variance was calculated with Formula 2, indicating how much variance cannot be attributed to any single factor and is jointly shared among factors.

$$R_{shared}^2 = R_{6factor}^2 - \sum_{i=1}^6 \Delta R_i^2$$

The residual variance indicates the variance that the model cannot explain, calculated as $1 - R_{6factor}^2$, and likely reflects biological noise or measurement error.. We included the variance explained by single factors, shared variance and residual for each soil responses in the supplementary information.'

We further included the table with R^2 values in the supplementary information (Table 2):

Table 2. Redundancy analysis (RDA) and variance partitioning to quantify the relative contribution of each factor and their interactions for all- group (independent R^2).

Response variable	Factor	Independent R^2
N-acetyl- β -glucosaminidase activity	Warming	0.159079
	Drought	0.396307
	Salinity	0.184495
	Surfactant	0.000441
	N deposition	0.002569
	Microplastic	0.000426
	Shared (interaction)	0.000000
	Residual	0.351265
	Warming	0.127104

β-D-glucosidase activity	Drought	0.075372
	Salinity	0.129810
	Surfactant	0.000000
	N deposition	0.000000
	Microplastic	0.000000
	Shared (interaction)	0.052782
	Residual	0.614933
	Warming	0.155493
	Drought	0.066334
	Salinity	0.276659
Phosphatase activity	Surfactant	0.000000
	N deposition	0.000000
	Microplastic	0.000000
	Shared (interaction)	0.120065
	Residual	0.381449
	Warming	0.008262
	Drought	0.015494
	Salinity	0.469099
	Surfactant	0.000396
	N deposition	0.002998
Decomposition	Microplastic	0.000844
	Shared (interaction)	0.398437
	Residual	0.104470
	Warming	0.000000
	Drought	0.443133
	Salinity	0.000000
	Surfactant	0.000000
	N deposition	0.000000
	Microplastic	0.000000
	Shared (interaction)	0.287355
Mean weight diameter	Residual	0.269512
	Warming	0.000000
	Drought	0.036772
	Salinity	0.313569
	Surfactant	0.000000
	N deposition	0.000000
	Microplastic	0.000000
	Shared (interaction)	0.086826
	Residual	0.562834
	Warming	0.031138
Water-stable aggregates	Drought	0.142639
	Salinity	0.066906
	Surfactant	0.012967
	N deposition	0.000000
	Microplastic	0.017452
	Shared (interaction)	0.261426
	Residual	0.467471
	Warming	0.002019
	Drought	0.000000
	Salinity	0.552363
Water drop penetration time	Surfactant	0.001124
	N deposition	0.000000
	Microplastic	0.000000
	Shared (interaction)	0.274997
Electrical conductivity		

pH	Residual	0.169497
	Warming	0.029258
	Drought	0.086963
	Salinity	0.072416
	Surfactant	0.125286
	N deposition	0.007198
	Microplastic	0.034358
	Shared (interaction)	0.000000
	Residual	0.908833
	Warming	0.000000
Phosphorus	Drought	0.000000
	Salinity	0.000000
	Surfactant	0.000000
	N deposition	0.0187398
	Microplastic	0.000000
	Shared (interaction)	0.000000
	Residual	1.0237610
	Warming	0.000000
	Drought	0.000000
	Salinity	0.003527902
Carbon-to-nitrogen ratio	Surfactant	0.000000
	N deposition	0.000000
	Microplastic	0.000000
	Shared (interaction)	0.000000
	Residual	1.051575794

We modified the results and discussion section accordingly. We added the overall values for shared variance and residuals in the figures as well.

5. Surfactant concentrations ("16 mg per kg soil") are not justified against real-world levels.

We have now provided a detailed explanation in the methods section regarding our choice of concentration. We now justified this as follows:

'Surfactant concentrations in graywater effluents can range from 0.7 to 70 mg L⁻¹ (Kuhnt, 1993), and from 0.2 to 20 g kg⁻¹ in sewage sludges, leading to soil concentrations of up to several mg kg⁻¹ (Wiel-Shafran et al., 2006). Experimental studies involving surfactant application have used between 1 and 6 different dose levels (Lehmann et al., 2023). A synthetic anionic surfactant was applied at concentrations of 16 mg kg⁻¹ in both a field study (Figge & Schöberl, cited in Ying, 2006) and a greenhouse experiment (Bi et al., 2024). Based on these findings, we selected this concentration of sodium dodecylbenzenesulfonate to simulate a surfactant-contaminated hotspot.'

6. In summary, the manuscript addresses an important topic in urban soil ecology but falls short of the high standards expected for publication.

Thank you for the valuable advice. We hope that our manuscript now meets the expectations for this journal. We believe our new drop-one experimental design could be a blueprint for many studies in urban ecology, and thus should be of interest to a broad audience in soil ecology and beyond.